# Evaluation of knowledge, attitude and practice towards cystic echinococcosis among undergraduate students in China

**Yijie Xu[1], Chengkai Luo[1], Jiacheng Liu[1], Congwei Shen[1], Xingming Ma[1,2]\***

**1** School of Health Management, Xihua University, Chengdu, China, **2** Health Promotion Center, Xihua University, Chengdu, China

\* ming2020xm@163.com

## Abstract

### Background

Cystic echinococcosis (CE), a zoonosis caused by *Echinococcus granulosus sensu lato* (EG), is a major public health burden in western China. The study aimed to evaluate the current status of KAP about CE among undergraduate students from western areas of China, and to provide a basis for developing health education strategies for college students on the prevention of CE.

### Methods

A cross-sectional study was conducted from September 2023 to April 2024 in China. A cluster sampling method was used to enroll participants of undergraduate students. The Cronbach's alpha coefficient of the questionnaire, the $\chi^2$ test for comparing rates, the independent sample T test for KAP scores, and linear regression analyses were conducted in the study.

### Results

A total of 724 students were included in the study. Approximately 65% (471) of the participants were female students and 35% (253) were male students, and most of them were non-medical (90%) students and from rural areas (60%). The total score of KAP was from 21 to 57, and the mean score of KAP was 41.92±5.78, and less than one fifth of undergraduate students had higher level knowledge (14.8%), positive attitudes (15.5%), and low-risk practices (8.6%). The education level was a significant factor in predicting knowledge, and upper-year students (junior/senior) had a higher level of knowledge about CE than lower-year students (OR = 1.59, p = 0.01). Both residence (OR = 0.57, p < 0.05) and knowledge (OR = 2.26, p<0.05) were significant predicting attitude factors, and students with more knowledge had a positive attitude. But the significant predictors of practice were gender (OR = 3.26, p < 0.05), education level (OR = 0.75, p < 0.05) and attitude (OR = 2.25, p = 0.05).

**Data availability statement:** The analysed data is all in the manuscript.

**Funding:** the Project of Sichuan University Student Ideological and Political Education Research Center (CSZ24049) and the Key Project for Education and Teaching Reform in Xihua University (XJJG2021040). The funders had no role in study design, data collection and analysis, decision to publish, or preparation of the manuscript.

**Competing interests:** The authors have declared that no competing interests exist.

## Conclusion

The KAP level and pass rate for knowledge of CE among Chinese students in western areas is low, and most participants showed insufficient or poor CE knowledge. It is necessary to strengthen the health education in school or by multimedia communication platforms on CE for college students in western areas and those with identified related factors.

## Introduction

Human and animal cystic echinococcosis (CE), a zoonotic disease caused by the larval stages of *Echinococcus granulosus sensu lato* (EG), is one of the neglected tropical diseases identified by the World Health Organization, receiving priority assistance for its prevention and management [1]. EG has a complex life cycle involving definitive hosts, intermediate hosts, and humans as dead-end hosts. Humans and/or intermediate hosts (primarily herbivores like sheep, etc.) are infected by ingesting the parasite's eggs in contaminated food and water, and the parasite then develops into larval stages in the intestines. Definitive hosts, typically domestic or wild canids, become infected by consuming the viscera of intermediate hosts that harbor the parasite larvae [2].

The pathological, epidemiological and geographical incidences of different echinococcus taxa vary widely [3]. CE is mainly prevalent in the western areas of China, such as the provinces of Xinjiang, Sichuan, Qinghai, Gansu, Ningxia, Tibet, and Inner Mongolia from which accounted for 98.38%, 98.43%, and 97.83% of cases in 2017, 2018, and 2019, respectively [4,5]. In hyperendemic regions, the infected shepherd dogs become the primary source of infection for both humans and livestock [1]. Between 2004 and 2017, EG infections showed an increasing trend, reaching a peak incidence of 0.3975 per 100,000 person-years in 2017 (National Population and Health Science Data Sharing Platform 2023), and most of them emerged in the last decade, and about 50 out of 100,000 people now suffer from the disease every year in endemic areas [5]. In western areas of socio-economic development and livestock production, CE remains a persistent problem.

CE is caused by human or animal ingestion of eggs of the genus EG, which spread between intermediate hosts of the animal and the final host through predator-prey interactions [6]. Humans become infected through contact with an infected final host or by eating fruit, vegetables or water contaminated with EG eggs, and the risk of disease is approximately 0.0002% [7,8]. Human infection with eggs can then lead to the development of echinococcosis cysts with a variety of complications and, if not treated in time, can seriously affect human health. Thus, CE is a public health problem that seriously threatens human health and restricts economic development [9].

Vaccines and treatments for CE remain challenging, as surgery is not suitable for all patients and drugs can cause serious adverse events and resistance [10]. To control CE effectively, it is essential to use the strategy of interrupting the transmission route to control the development of the parasite at the stages of its life cycle. In many countries and regions, including China, it is important to note that the dogs are the primary source of infection for both humans and livestock. Dogs are infected by eating into the viscera of sick animals, and the eggs discharged by sick dogs pollute the natural environment, such as pasture, water source and animal products such as wool [6,7]. Personal and food hygiene, dogs and dog food in areas with a high incidence of the disease are managed and controlled to reduce the likelihood of them becoming hosts.

Inadequate knowledge of disease prevention and ignorance of preventive practices contribute to some extent to the failure to control cystic echinococcosis in its early stages. It is

important to understand the knowledge, attitudes, and practices (KAP) of college students about echinococcosis for the prevention and control of cystic echinococcosis [11]. Particularly for students from western areas of China where the incidence of cystic echinococcosis is high, it is valuable if health education can be effectively accepted and thus not only promote changes in behaviour with positive effects, but also be disseminated to families and family-based communities [12].

The previous KAP studies showed that the incidence of echinococcosis remains high in economically underdeveloped countries such as Morocco, Peru, Uganda, Pakistan, Iraq, Sudan, Morocco, Iran and Algeria. The majority of participants showed poor practices in relation to this disease, increasing the chances of further spread of the parasite circulation in the animal and human populations at risk [13–21]. But in China, the previous KAP studies from 2013 to 2020 mainly focused on the understanding of echinococcosis control knowledge among residents and primary and middle school students in the provinces of Gansu [22], Qinghai [23,24], Sichuan [25,26] and Tibetan Autonomous Region [10,27]. The survey of local residents and primary and secondary school students showed that the pass rate for knowledge of echinococcosis ranged from 6% to 95%, demonstrating that the prevalence of the disease is related to a lack of knowledge, awareness and poor hygiene practices in the local community.

Although the aforementioned studies have analyzed the KAP characteristics of populations related to echinococcosis, these studies are limited in terms of knowledge coverage and respondents. Fewer investigations have been conducted to examine the KAP of university students in western areas of China related to cystic echinococcosis. Therefore, the aim of the study was to assess the current cystic echinococcosis knowledge, attitudes and practices of Chinese college students in western areas. The results can be used to develop successful cystic echinococcosis health education strategies for the university students and will provide important insights into current KAP levels among students.

## Materials and methods

### Study design

To assess college students' understanding, attitudes, and behaviors (KAP) regarding cystic echinococcosis, a cross-sectional survey was carried out in China from the fall to spring semester, spanning September 1, 2023 to April 30, 2024. The study adhered to the STROBE checklist for observational studies (S1 Appendix in S1 File).

### Study population

A cluster random sampling approach was employed to recruit participants at Xihua University. Students were selected from three hundred eligible classes, and then fourteen classes were randomly selected using the excel random function sampling method. Next, 769 students from western China voluntarily participated in this cross-sectional study, regardless of grades and majors, including both medical and non-medical students.

### Sample size calculation

A sample size of participants was calculated using the online random sampling formula (https://tool.iikx.com/sample/) and n = $Z^2$*p*(1 - p)/ $e^2$. The test significance level was set at α = 0.05, allowable marginal error $e$ = 4%, and the cognitive eligibility rate was reported as 33% in the literature [25], and a minimum sample size of participants was 531. Considering potential dropouts, the final sample size of participants was determined to be 796 (1.5 times).

### Ethics

The present study adhered to the principles outlined in the Declaration of Helsinki, and approval for the study was obtained from the curriculum development committee at the School of Health Management of our university (XJJG2021040-40). We ensured confidentiality for all respondents and stressed the voluntary aspect of the study. Participants were guaranteed that their data would solely be utilized for research purposes. Both the informed consent form and the questionnaire were designed to be fully anonymous, and informed consent was obtained from all participants.

### Questionnaire design

A questionnaire used to gather data on KAP was based on a literature review of previous studies [25,28,29]. The questionnaire included all 37 items, which were divided into five domains (S2 Appendix in S1 File). Meanwhile, we discussed with a Chinese expert together to confirm that the version of questionnaire was culturally appropriate for Chinese students. Next, a pre-survey of questionnaire was conducted and 31 sample were collected, and then the Cronbach's α coefficients were analyzed. The Cronbach's alpha coefficients of questionnaire were 0.72, which indicated a good internal consistency and could be used for formal trials.

The first domain with seven questions collected sociodemographic data from participants. The sociodemographic variables included gender, age, nationality, educational level, monthly living expenses, and residence.

In the cystic echinococcosis knowledge survey, there were eleven questions with "True" or "False" options, where "True" denoted a correct response indicating familiarity with echinococcosis knowledge, and a correct response got 1 point for a total of 11 points. In the 11-question survey, knowledge level was classified into three categories as poor ( < 5 points, < 50%), insufficient (5–8 points, 50%–75%), and good (> 8 points, >75%) knowledge levels, respectively.

In the domain of attitude towards cystic echinococcosis, there were eight questions with "yes" or "no" choices, where "yes" represented a positive response indicating a favorable attitude and a positive attitude got 1 point for a total of 8 points. In the 8-question item, attitude level was classified into three categories as negative ( < 4 points, < 50%), neutral (4–6 points, 50%–75%), and positive (> 6 points, >75%) attitude levels, respectively.

The behavior domain investigation related to cystic echinococcosis included ten Likert-type statements assessing participants' practices concerning echinococcosis. The scale progression from "never", "sometimes", "neutral", "often" to "always" reflected the degree of positive practice. The score ranged from one (never), two (sometimes), three (neutral), four (often) to five (always) for items of 1, 2, 4, 6, 7, and 8, while the score assigned 5 to never, 4 to sometimes, 3 to neutral, 2 to often, 1 to always for items of 3, 5, 9, and 10, and then total practice scores were calculated. In the 10-question scale, practice level was classified into three levels as high-risk ( < 25 points, < 50%), moderate-risk (25–37 points, 50–75%), and low-risk (>37 points, >75%) practice levels, respectively.

In investigating the ways of health education about echinococcosis knowledge, we collected different channels through which college students acquired knowledge about cystic echinococcosis, including: "Community doctor health propaganda", "Multimedia communication platforms such as television, radio, internet, WeChat", "Promotional display media like billboards, posters, brochures", "Family or friends", "School health education", and other ways.

### Statistical analysis

Data analysis was performed using IBM SPSS Statistics version 27 (IBM Corp., Armonk, NY, USA). Categorical variables were reported as frequencies and percentages, while

students' scores of KAP (knowledge, attitude, and practice) are reported as means and standard deviations (SD). Cronbach's α was utilized to assess the internal consistency of the questionnaires (attitudes and practices domains), and 0.6 or higher is an acceptable level. The Chi-square test was used to analyze the differences in categorical variables, and the independent sample T test was used to analyze the continuous variable differences between socio-demographic characteristics. Logistic regression analysis was used to identify associated variables with good knowledge and positive attitude as well as lower-risk practice. The included variables were age, gender, nationality, grade level, monthly living expenses, type of specializations, and residence. Univariate analysis was performed in the binary logistic regressions.

## Results

### Socio-demographic characteristics of the study population

The results of Cronbach alpha coefficient in the study indicated a good internal consistency, with coefficients of 0.81, 0.60, and 0.76 for the knowledge, attitude, and practice scales, respectively.

In the study, a total of 730 questionnaires were collected from 796 undergraduate students, for a response rate of 92% (all data in S3 Appendix in S1 File). After excluding 6 incomplete ones, 724 questionnaires were included, resulting in an effective recovery rate of 99%. Among them, 471 (65.1%) students were female, while 253 (34.9%) students were male. The age of the great majority of (79.8%) students was < 20 years, and the age of the interviewed subjects ranged from 17 to 23 years old. The nationality of the majority of enrolled students was Han Chinese (92.5%), and most participants were the lower-year undergraduates (freshmen and sophomores, 76.9%). About nine out of ten participants (89.5%) were non-medical students, and nine out of ten students (88.3%) have a monthly living allowance of no more than 2,000 yuan. Additionally, 432 students (59.7%) came from rural areas and 292 students (40.3%) from urban areas (Table 1).

The total score of KAP was from 21 to 57, and the mean score of KAP was 41.92±5.78 out of 69. Of the 724 participants, only 41 students (5.7%) had a high KAP level and scored over 51 points (>75%). The study findings show a significant difference in KAP score based on gender and residence (p < 0.05) (Table 1).

### Knowledge about echinococcosis

Of the 724 participants, only 14.8% (n = 107) had a good knowledge level, and half of the participants (49.7%) showed that CE knowledge was insufficient (Fig 1 and Table 2). The mean score of knowledge was 4.34±3.50 out of 11 items. The mean score of the echinococcosis transmission knowledge was 2.38±2.16 out of 7 items, while the mean score of the echinococcosis prevention knowledge was 1.96±1.93 out of 4 items. Most of the participants showed poor knowledge of echinococcosis transmission and prevention (Fig 1).

In the demographic characteristics variable comparison, there was an obvious difference in students' CE knowledge in terms of education level (Table 2 and Table 3). However, the mean score of echinococcosis transmission knowledge in the female students (2.52 ± 2.11) was noticeably higher than that of the male students (2.11± 2.22, p < 0.05). The mean score of CE prevention knowledge in the upperclassmen (junior/senior) (2.60± 2.12) and medical students (2.51± 2.21) was visibly higher than that in the underclassmen (freshman/sophomore) (1.77 ± 1.83) and non-medical students (1.95± 1.92) (p < 0.05) (Table 3). However, there was no significant difference in students' CE knowledge in terms of other variables (p>0.05).

**Table 1. Demographic characteristics and KAP score analysis of the participants.**

| Variables | Proportion (n = 724) | Frequency (%) | Scores of KAP (mean±SD) | P value |
|---|---|---|---|---|
| **Gender** | | | | |
| Male | 253 | 34.9 | 41.07±5.75 | 0.004 |
| Female | 471 | 65.1 | 42.38±5.76 | |
| **Age, year** | | | | |
| 17 ～ 20 years | 578 | 79.8 | 42.08±5.64 | 0.13 |
| ≥20 years | 146 | 20.2 | 41.28±6.31 | |
| **Nationality** | | | | |
| Han Chinese | 670 | 92.5 | 41.91±5.72 | 0.89 |
| Ethnic minority | 54 | 7.5 | 42.02±6.54 | |
| **Grade level** | | | | |
| Lower-year undergraduate | 557 | 76.9 | 41.80±5.73 | 0.31 |
| Upper-year undergraduates | 167 | 23.1 | 42.32±5.91 | |
| **Monthly living expenses** | | | | |
| ≤2000 yuan | 640 | 88.3 | 41.98±5.91 | 0.46 |
| >2000 yuan | 84 | 11.7 | 41.49±4.68 | |
| **Type of specializations** | | | | |
| Non-medical specialty | 648 | 89.5 | 41.79±5.80 | 0.07 |
| Medical specialty | 76 | 10.5 | 43.04±5.58 | |
| **Residence** | | | | |
| Urban area | 292 | 40.3 | 42.68±5.55 | 0.004 |
| Rural area | 432 | 59.7 | 41.41±5.88 | |

KAP (knowledge, attitude, and practice).

## Attitude toward echinococcosis

Of the 724 participants, only 15.5% (n = 112) had a positive attitude level, and most participants (78.2%, n = 566) had a neutral attitude level towards CE (Fig 1 and Table 4). There were significant differences in the variables of gender and place of residence. More male than female students had a positive attitude towards echinococcosis, and more students from urban areas than from rural areas had a positive attitude towards echinococcosis. Furthermore, the mean attitude score was 5.32±1.37 out of 8. When comparing the attitude scores of the demographic characteristics variables, the mean score of attitude towards CE in medical students (5.61±1.39) was significantly higher than that of non-medical students (5.28±1.36) (p = 0.05). Students from urban areas had a more favourable attitude towards echinococcosis (p < 0.05). However, there was no significant difference in students' attitudes towards CE with respect to other variables (p>0.05).

## Practice toward echinococcosis

Of the 724 participants, only 8.6% (n = 62) had a better level of practice with low risk, and more than 90.2% of students had a moderate level of practice with moderate risk (Fig 1 and Table 5). There were significant differences in the variables of gender and residence. The mean practice score was 32.26±3.38 out of 50. Furthermore, when comparing the practice scores in the demographic characteristic variable, the mean practice score for CE in the female students (32.47±3.68) was significantly higher than that of the male students (31.87±2.72) (p < 0.05). However, there was no significant difference in students' practices towards CE in terms of other variables (p>0.05).

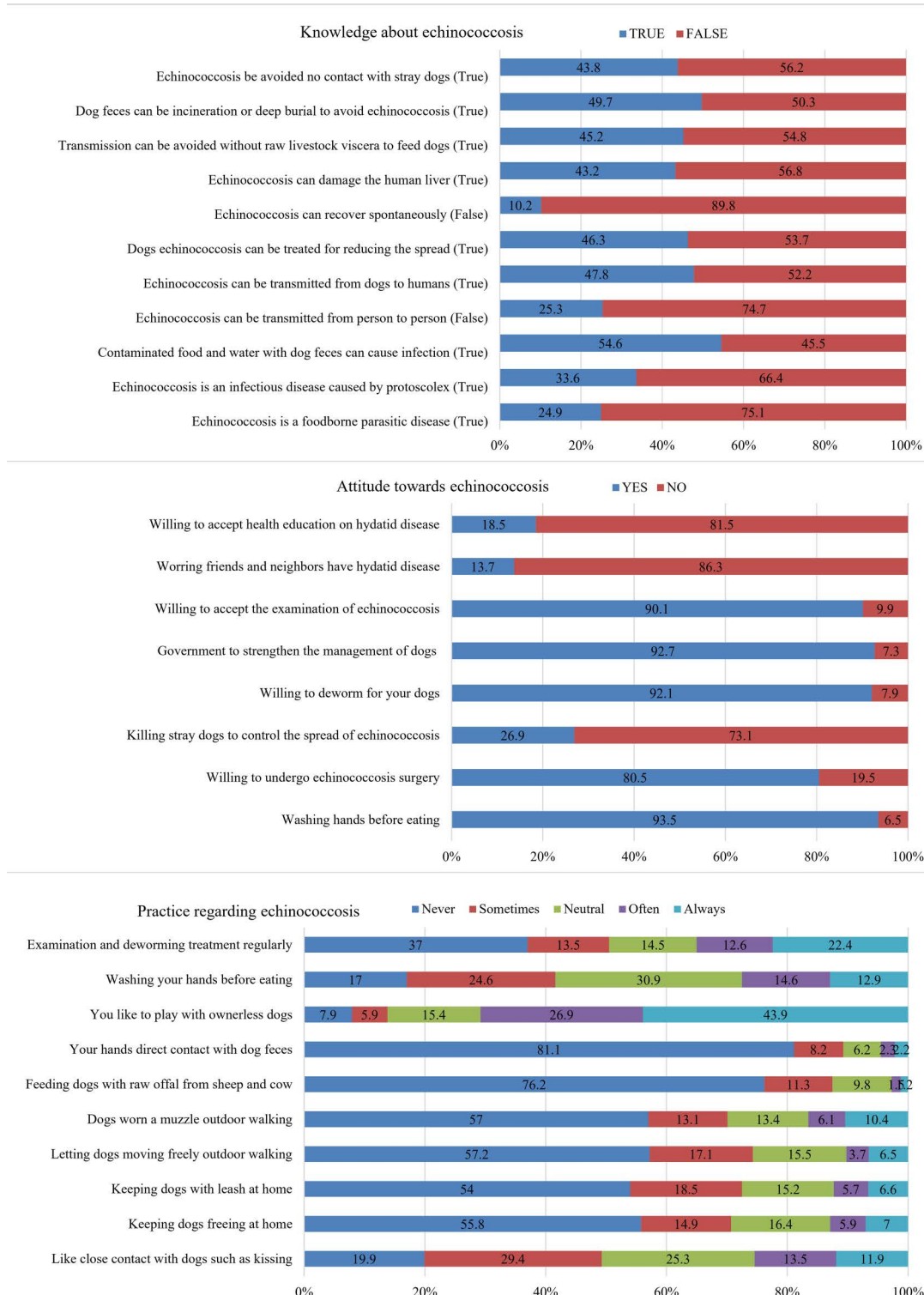

**Fig 1. Responses to questions related to AKP (knowledge, attitude, and practice) about cystic echinococcosis (n = 724).**

**Table 2. The frequency of knowledge response among students regarding cystic echinococcosis (n = 724).**

| Variables(n) | Poor knowledge n (%) | Insufficient knowledge n (%) | Good knowledge n (%) | P value |
|---|---|---|---|---|
| **Gender** | | | | |
| Male (253) | 140(55%) | 77(30%) | 36(14%) | 0.06 |
| Female (471) | 220(47%) | 180(38%) | 71(15%) | |
| **Age, year** | | | | |
| 17 ~ 20 years (578) | 281(49%) | 218(38%) | 79(13%) | 0.03 |
| ≥20 years (146) | 79(54%) | 39(27%) | 28(19%) | |
| **Nationality** | | | | |
| Han Chinese (670) | 335(50%) | 235(35%) | 100(15%) | 0.69 |
| Ethnic minority (54) | 25(46%) | 22(40%) | 7(13%) | |
| **Grade level** | | | | |
| Lower-year undergraduate (557) | 290(52%) | 193(35%) | 74(13%) | 0.03 |
| Upper-year undergraduates (167) | 70(42%) | 64(38%) | 33(20%) | |
| **Monthly living expenses** | | | | |
| ≤2000 yuan (640) | 311(49%) | 233(36%) | 96(15%) | 0.23 |
| >2000 yuan (84) | 49(58%) | 24(29%) | 11(13%) | |
| **Type of specializations** | | | | |
| Non-medical specialty (648) | 325(50%) | 230(36%) | 93(14%) | 0.60 |
| Medical specialty (76) | 35(46%) | 27(36%) | 14(18%) | |
| **Residence** | | | | |
| Urban area (292) | 133(46%) | 115(39%) | 44(15%) | 0.14 |
| Rural area (432) | 227(53%) | 142(33%) | 63(14%) | |

Data are presented as frequency (n) and percentage (%) of knowledge response. The Chi-square test was used for the knowledge response in the analysis. Statistical significance was set at a level of $P < 0.05$.

## Assessment of the variables associated with KAP levels

The results of the binary logistic regressions showed that both gender and grade level were significant factors in predicting knowledge, attitudes and practice (Table 6). Specifically, the significant predictors of knowledge were grade level (OR = 1.59, $p = 0.01$). Upper-year students (junior/senior) were 1.59 times more likely to have a high level of knowledge about CE than lower-year students (freshman/sophomore) ($p < 0.05$). The significant predictors of attitude were gender (OR = 0.56, $p < 0.05$), residence (OR = 0.57, $p < 0.05$) and knowledge (OR = 2.26, $p < 0.05$). Students with a high level of knowledge were significantly more likely to have a high level of attitude compared to those with a low level of knowledge. Conversely, being a female student significantly reduced the likelihood of having a high level of attitude compared to male students. The significant predictors of practice were gender (OR = 3.26, $p < 0.05$), education level (OR = 0.75, $p < 0.05$) and attitude (OR = 2.25, $p = 0.05$). Students with a high level of attitude were significantly more likely to have a low risk of practice compared to those with a low level of attitude.

## Health education approach toward echinococcosis

The results showed that the highest frequency of obtaining CE knowledge was through multimedia communication platforms, with 48.2% of students, followed by school health education with 42.5% (Fig 2A). In contrast, community doctor health propaganda was the

**Table 3. Awareness scores of cystic echinococcosis-related knowledges among participants (n = 724).**

| Variables (n) | Scores of transmission (mean±SD) | P value | Scores of prevention (mean±SD) | P value | Total Scores of knowledges (mean±SD) | P value |
|---|---|---|---|---|---|---|
| **Gender** | | | | | | |
| Male (253) | 2.11±2.22 | 0.01 | 1.78±1.94 | 0.06 | 3.89±3.66 | 0.01 |
| Female (471) | 2.52±2.11 | | 2.06±1.90 | | 4.59±3.40 | |
| **Age, year** | | | | | | |
| 17 ~ 20 years (578) | 2.44±2.14 | 0.11 | 1.96±1.92 | 0.99 | 4.41±3.45 | 0.32 |
| ≥20 years (146) | 2.12±2.21 | | 1.97±2.00 | | 4.09±3.72 | |
| **Nationality** | | | | | | |
| Han Chinese (670) | 2.38±2.17 | 0.91 | 1.97±1.95 | 0.82 | 4.34±3.53 | 0.95 |
| Ethnic minority (54) | 2.41±1.99 | | 1.91±1.73 | | 4.31±3.21 | |
| **Grade level** | | | | | | |
| Lower-year undergraduate (557) | 2.42±2.21 | 0.36 | 1.77±1.83 | 0.001 | 4.19±3.48 | 0.03 |
| Upper-year undergraduates (167) | 2.25±1.97 | | 2.60±2.12 | | 4.85±3.56 | |
| **Monthly living expenses** | | | | | | |
| ≤2000 yuan (640) | 2.42±2.17 | 0.16 | 1.95±1.92 | 0.71 | 4.37±3.53 | 0.51 |
| >2000 yuan (84) | 2.07±2.04 | | 2.04±2.05 | | 4.11±2.28 | |
| **Type of specializations** | | | | | | |
| Non-medical specialty (648) | 2.39±2.18 | 0.79 | 1.90±1.89 | 0.009 | 4.29±3.50 | 0.20 |
| Medical specialty (76) | 2.32±1.95 | | 2.51±2.21 | | 4.83±3.52 | |
| **Residence** | | | | | | |
| Urban area (292) | 2.53±2.17 | 0.11 | 2.13±1.98 | 0.05 | 4.66±3.45 | 0.04 |
| Rural area (432) | 2.28±2.15 | | 1.85±1.89 | | 4.13±3.53 | |

The independent sample t-test was used for the scores of cystic echinococcosis-related knowledge in the analysis. Statistical significance was set at a level of P < 0.05.

least popular method, with only 16.6% of students reporting it. About 18% of the university students learned about echinococcosis in one way, while 9.7% learned about echinococcosis in five ways (Fig 2B). Particularly, multimedia communication platforms and school health education were found to be the most effective ways for students to acquire cystic echinococcosis knowledge.

In the variable comparison of gender, a higher percentage of female students (51%) obtained echinococcosis knowledge from multimedia communication platforms compared to male students (47%). Similarly, a higher percentage of male students (48%) obtained echinococcosis knowledge through school health education compared to female students (32%) (p < 0.05). However, there were no significant differences in the ways students obtained health knowledge based on other demographic characteristics variables (p>0.05) (Table 7).

## Discussion

Undergraduate students represent a significant portion of the young population and can be key agents in spreading health information about CE, thereby contributing to raising awareness and promoting behavioural change within their households. Among the 724 participants, the findings revealed that less than one-fifth of undergraduate students had higher level knowledge, positive attitudes, and low-risk practices, indicating that the current overall CE health literacy needs to be improved. Urban students exhibited higher KAP scores compared to rural counterparts, and disparities in knowledge, attitude, and practices based on gender and residence. Multimedia communication platforms and school health education were the

**Table 4. The frequency and scores of cystic echinococcosis-related attitude response among students regarding echinococcosis (n = 724).**

| Variables (n) | Negative attitude n (%) | Neutral attitude n (%) | Positive attitude n (%) | P value | Scores of attitudes (mean±SD) | P value |
|---|---|---|---|---|---|---|
| **Gender** | | | | | | |
| Male (253) | 30(12%) | 160(63%) | 63(25%) | 0.001 | 5.31±1.69 | 0.95 |
| Female (471) | 16(3%) | 406(86%) | 49(10%) | | 5.32±1.16 | |
| **Age, year** | | | | | | |
| 17 ~ 20 years (578) | 36(6%) | 459(79%) | 83(14%) | 0.233 | 5.29±1.37 | 0.35 |
| ≥20 years (146) | 10(6%) | 107(72%) | 29(20%) | | 5.41±1.36 | |
| **Nationality** | | | | | | |
| Han Chinese (670) | 40(6%) | 529(79%) | 101(15%) | 0.15 | 5.32±1.35 | 0.60 |
| Ethnic minority (54) | 6(11%) | 37(69%) | 11(20%) | | 5.22±1.59 | |
| **Grade level** | | | | | | |
| Lower-year undergraduate (557) | 36(6%) | 442(79%) | 79(14%) | 0.21 | 5.29±1.33 | 0.43 |
| Upper-year undergraduates (167) | 10(6%) | 124(74%) | 33(20%) | | 5.39±1.48 | |
| **Monthly living expenses** | | | | | | |
| ≤2000 yuan (640) | 41(6%) | 499(78%) | 100(16%) | 0.93 | 5.32±1.36 | 0.97 |
| >2000 yuan (84) | 5(6%) | 67(80%) | 12(14%) | | 5.32±1.41 | |
| **Type of specializations** | | | | | | |
| Non-medical specialty (648) | 42(6%) | 512(79%) | 94(15%) | 0.11 | 5.28±1.36 | 0.05 |
| Medical specialty (76) | 4(5%) | 54(71%) | 18(24%) | | 5.61±1.39 | |
| **Residence** | | | | | | |
| Urban area (292) | 9(3%) | 228(78%) | 55(19%) | 0.003 | 5.56±1.17 | 0.001 |
| Rural area (432) | 37(9%) | 338(78%) | 57(13%) | | 5.15±1.47 | |

Data are presented as frequency (n) and percentage (%) of attitude response. The Chi-square test was used for the attitude response in the analysis. The independent sample t-test was used for the scores of cystic echinococcosis-related attitudes in the analysis. Statistical significance was set at a level of P < 0.05.

most effective ways for students to acquire echinococcosis knowledge, indicating the importance of utilizing diverse educational channels for dissemination.

The studies have shown that EG infections are dependent on variables such as health knowledge and personal hygiene, which facilitate close contact with tapeworm eggs. The more people know about the causes, symptoms, transmission, prevention and treatment of CE, the more positive attitudes and behaviors towards CE prevention they have, and the lower the risk of infection of EG. [30,31]. In the study, the passing rate of the CE knowledge is 50%. Only 14.8% of students demonstrated a good level of knowledge, and most of the participants showed that their CE knowledge was insufficient or poor. Particularly in the question of "CE is a foodborne parasitic disease and an infectious disease caused by protoscolex", only 25%-34% of students chose the correct answer, showing that most students do not understand how CE is caused. Similarly, a previous study in Sichuan Province showed that middle school students had some deficiencies in their knowledge of echinococcosis and did not understand the basic principles [11]. On the contrary, for the question "CE can recover spontaneously and be transmitted from person to person", only a small number of students (10%-25%) believe that the knowledge description was correct, showing that most students still have some understanding of the transmission of CE. The answers to other questions indicated that students do not have a high level of understanding of CE and that their knowledge was inadequate or even poor.

Differences were found by level of education, gender and subject, with female students, upper-year students and medical students having a high level of knowledge about CE. Female

**Table 5. The frequency of practice response among students regarding cystic echinococcosis (n = 724).**

| Variables (n) | High-risk practice n (%) | Moderate-risk practice n (%) | Low-risk practice n (%) | P value | Scores of practices (mean±SD) | P value |
|---|---|---|---|---|---|---|
| **Gender** | | | | | | |
| Male (253) | 3(1%) | 240(95%) | 10(4%) | 0.005 | 31.87±2.72 | 0.02 |
| Female (471) | 6(1%) | 413(88%) | 52(11%) | | 32.47±3.68 | |
| **Age, year** | | | | | | |
| 17～20 years (578) | 6(1%) | 522(90%) | 50(9%) | 0.60 | 32.38±3.30 | 0.05 |
| ≥20 years (146) | 3(2%) | 131(90%) | 12(8%) | | 31.78±3.68 | |
| **Nationality** | | | | | | |
| Han Chinese (670) | 7(1%) | 609(91%) | 54(8%) | 0.05 | 32.24±3.26 | 0.62 |
| Ethnic minority (54) | 2(4%) | 44(81%) | 8(15%) | | 32.48±4.66 | |
| **Grade level** | | | | | | |
| Lower-year undergraduate (557) | 5(1%) | 502(90%) | 50(9%) | 0.24 | 32.32±3.39 | 0.42 |
| Upper-year undergraduates (167) | 4(2%) | 151(90%) | 12(8%) | | 32.08±3.37 | |
| **Monthly living expenses** | | | | | | |
| ≤2000 yuan (640) | 9(1%) | 572(89%) | 59(10%) | 0.11 | 32.29±3.47 | 0.56 |
| >2000 yuan (84) | 0(0%) | 81(96%) | 3(4%) | | 32.06±2.67 | |
| **Type of specializations** | | | | | | |
| Non-medical specialty (648) | 8(1%) | 583(90%) | 57(9%) | 0.80 | 32.22±3.42 | 0.35 |
| Medical specialty (76) | 1(1%) | 70(92%) | 5(7%) | | 32.61±3.09 | |
| **Residence** | | | | | | |
| Urban area (292) | 1(1%) | 259(87%) | 32(12%) | 0.03 | 32.46±3.35 | 0.20 |
| Rural area (432) | 8(2%) | 394(91%) | 30(7%) | | 32.13±3.40 | |

Data are presented as frequency (n) and percentage (%) of practice response. The Chi-square test was used for the practice response in the analysis. The independent sample t-test was used for the scores of cystic echinococcosis-related practices in the analysis. Statistical significance was set at a level of P < 0.05.

students, upper-year students, and medical students may have more opportunities to learn knowledge of CE through their curriculum or additional studies [5,32]. Therefrom, it is important to enhance the awareness of CE among college students to improve their understanding of this disease.

Fewer students in the study had negative attitudes (6%) and high-risk behaviors (1%) of CE prevention and treatment, with notable differences based on gender and urban/rural background. There were three attitude-related questions with a very low correct response rate of about 20%. Most of the questions were answered in the affirmative, with more than 90% of the students with a positive attitude. with notable differences based on gender and urban/rural backgrounds. Female students and urban dwellers exhibited more favorable attitudes, highlighting the influence of gender and environmental factors on attitudes. A study at the University of Algeria showed that students in different regions also received different levels of health education [33].

Most students (90%) showed moderate-risk practice levels, with gender and urban/rural influencing practical behavior significantly. Female and urban students displayed better practices than males and rural students, underlining gender and residence disparities in CE management behaviors. Females were at higher risk of echinococcosis infection than males, mainly because they were more involved in most family chores and activities, such as feeding dogs and preparing food for their families [5]. Women are often seen as family health managers, and society expects them to actively seek disease prevention information, while men

**Table 6. The binary regressions of knowledge, attitudes, and practice levels (n = 724).**

| | Knowledge level | | | | Attitudes level | | | | Practice level | | | |
|---|---|---|---|---|---|---|---|---|---|---|---|---|
| | | 95% CI | | | | 95% CI | | | | 95% CI | | |
| | OR | Lower | Upper | P value | OR | Lower | Upper | P value | OR | Lower | Upper | P value |
| **Gender** | | | | | | | | | | | | |
| Females VS Males (parameter = 0) | 1.17 | 0.81 | 1.67 | 0.39 | 0.77 | 0.56 | 1.06 | 0.04 | 3.26 | 1.90 | 5.61 | < 0.01 |
| **Age, year** | | | | | | | | | | | | |
| ≥20 years VS 17~20 years (parameter = 0) | 0.81 | 0.51 | 1.29 | 0.38 | 1.16 | 0.77 | 1.75 | 0.45 | 1.21 | 0.69 | 2.12 | 0.49 |
| **Nationality** | | | | | | | | | | | | |
| Ethnic minority VS Han Chinese (parameter = 0) | 0.58 | 0.27 | 1.22 | 0.15 | 1.19 | 0.67 | 2.12 | 0.53 | 1.72 | 0.84 | 3.49 | 0.13 |
| **Grade level** | | | | | | | | | | | | |
| Upper-year undergraduate VS Lower-year undergraduates (parameter = 0) | 1.59 | 1.05 | 2.41 | 0.03 | 1.12 | 0.76 | 1.65 | 0.55 | 0.75 | 0.42 | 1.29 | 0.04 |
| **Monthly living expenses** | | | | | | | | | | | | |
| >2000 yuan VS ≤2000 yuan (parameter = 0) | 0.50 | 0.27 | 0.93 | 0.03 | 1.02 | 0.61 | 1.66 | 0.95 | 0.54 | 0.25 | 1.19 | 0.13 |
| **Type of specializations** | | | | | | | | | | | | |
| Medical VS Non-medical (parameter = 0) | 1.35 | 0.76 | 2.38 | 0.29 | 1.06 | 0.62 | 1.79 | 0.82 | 1.31 | 0.63 | 1.74 | 0.46 |
| **Residence** | | | | | | | | | | | | |
| Cities/towns VS Rural area (parameter = 0) | 0.73 | 0.51 | 1.03 | 0.08 | 0.57 | 0.41 | 0.78 | < 0.01 | 0.88 | 0.56 | 1.39 | 0.60 |
| **Knowledge level** | | | | | | | | | | | | |
| High VS Low (parameter = 0) | – | – | – | – | 2.26 | 1.59 | 3.21 | < 0.01 | 1.31 | 0.82 | 2.09 | 0.24 |
| **Attitudes** | | | | | | | | | | | | |
| High VS Low (parameter = 0) | – | – | – | – | – | – | – | – | 2.28 | 1.47 | 3.55 | < 0.01 |

OR = Odds Ratio. 95% CI = 95% Confidence interval. Lower-year undergraduates (freshman and sophomore), and upper-year undergraduates (junior and senior). "-" Not included. The $R^2$ for the level of knowledge, attitudes and practice were 0.03 (Hosmer and Lemeshow test: $\chi^2$ = 8.00, P = 0.33), 0.07 (Hosmer and Lemeshow test: $\chi^2$ = 12.60, P = 0.13), and 0.10 (Hosmer and Lemeshow test: $\chi^2$ = 8.61, P = 0.37), respectively. The frequency of level was calculated based on the knowledge with high levels (good) and low levels (poor and insufficient), attitude with high levels (positive) and low levels (negative and neutral), and practice with high levels (low-risk) and low levels (high-risk and moderate-risk), respectively.

may ignore such knowledge through role labels [5]. To reduce the risk of CE, female students tended to acquire more health knowledge about the echinococcosis disease and adopt health behaviors. Similarly, urban students, benefiting from greater access to educational opportunities and resources, were more likely to gain comprehensive knowledge about CE and implement preventive health practices. Similarly, urban students, benefiting from greater access to educational opportunities and resources, were more likely to gain comprehensive knowledge about CE and implement preventive health practices. There are also differences in the health ecology of communities. Urban communities tend to have better food quarantine systems and medical advice channels to support preventive behaviour. Such resources are difficult to put into practice in rural areas, even where knowledge exists. In addition, rural families may put disease prevention on the backburner due to economic pressures, which indirectly affects students' value perception of health behaviours.

While dogs, especially sheepdogs, are the most common definitive hosts for CE, they are also transmitters of echinococcosis [34]. More and more people see dogs as close friends, and dogs have a higher status in the human family [35]. Although about 80% of the students are correct in the behavior of never directly touching dog faeces with their hands and never feeding dogs raw offal from cattle and sheep, half of the students are incorrect in the behavior of never keeping dogs with leashes in the yard, and keeping the dog on a leash while walking outdoors. Therefore, it is important to pay attention to the problems of dogs in the transmission of CE.

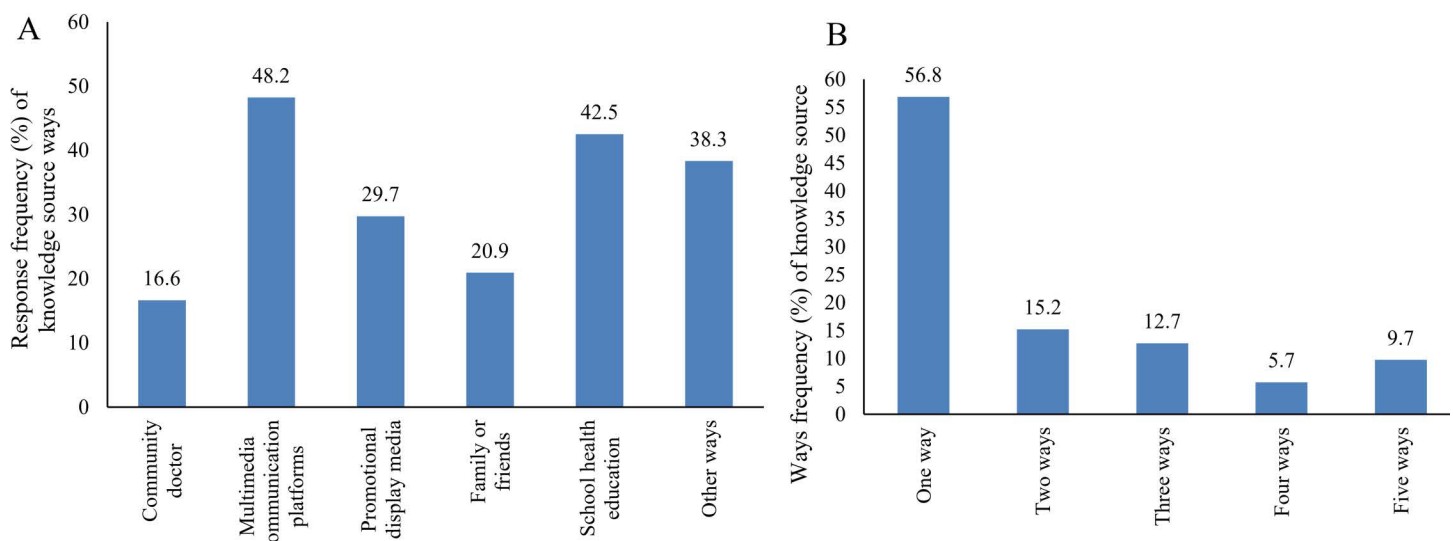

**Fig 2. Responses frequency to knowledge source about cystic echinococcosis (A: Response frequency on knowledge source ways, B: Ways frequency of knowledge source).** The health education ways of echinococcosis knowledge include community doctor health propaganda, multimedia communication platforms (television, radio, internet, WeChat), promotional display media (billboards, posters, brochures), family or friends, school health education and other ways.

**Table 7. The ways of health education about cystic echinococcosis knowledge among participants (n = 724).**

| Variables (n) | Community doctor | Multimedia communication platform | Promotional display media | Family or friends | School health education | Other ways | P value |
|---|---|---|---|---|---|---|---|
| **Gender** | | | | | | | |
| Male (253) | 58(23%) | 129(51%) | 73(29%) | 64(25%) | 81(32%) | 103(41%) | 0.001 |
| Female (471) | 62(13%) | 220(47%) | 142(30%) | 87(18%) | 227(48%) | 175(37%) | |
| **Age, year** | | | | | | | |
| 17 ~ 20 years (578) | 102(18%) | 297(51%) | 178(31%) | 120(21%) | 257(44%) | 222(38%) | 0.53 |
| ≥20 years (146) | 18(12%) | 52(36%) | 37(25%) | 31(21%) | 51(35%) | 56(38%) | |
| **Nationality** | | | | | | | |
| Han Chinese (670) | 109(16%) | 326(49%) | 196(29%) | 140(21%) | 289(43%) | 255(38%) | 0.15 |
| Ethnic minority (54) | 11(20%) | 23(43%) | 19(35%) | 11(20%) | 11(20%) | 23(43%) | |
| **Grade level** | | | | | | | |
| Lower-year undergraduates (557) | 95(17%) | 281(50%) | 173(31%) | 112(20%) | 239(43%) | 106(19%) | 0.001 |
| Upper-year undergraduates (167) | 25(15%) | 68(41%) | 42(25%) | 39(23%) | 69(41%) | 124(74%) | |
| **Monthly living expenses** | | | | | | | |
| ≤2000 yuan (640) | 106(17%) | 324(51%) | 199(31%) | 135(21%) | 287(45%) | 242(38%) | 0.06 |
| >2000 yuan (84) | 14(17%) | 25(30%) | 16(19%) | 16(19%) | 21(25%) | 36(43%) | |
| **Type of specializations** | | | | | | | |
| Non-medical specialty (648) | 103(16%) | 304(47%) | 191(29%) | 137(21%) | 273(42%) | 250(39%) | 0.37 |
| Medical specialty (76) | 17(22%) | 45(59%) | 24(32%) | 14(18%) | 35(46%) | 22(29%) | |
| **Residence** | | | | | | | |
| Urban area (292) | 46(16%) | 153(52%) | 83(28%) | 66(23%) | 128(44%) | 102(35%) | 0.46 |
| Rural area (432) | 74(17%) | 196(45%) | 132(31%) | 85(20%) | 180(42%) | 176(41%) | |

Data are presented as frequency (n) and percentage (%) of health education way response. The Chi-square test was used for the health education response in the analysis. Statistical significance was set at a level of P < 0.05.

Improving the KAP level remains one of the most effective strategies for helping control and prevent echinococcosis. All three elements are interdependent based on the relationships that exist between knowledge, attitude, and behavior; knowledge influences attitude, attitude influences behavior, and knowledge indirectly influences behavior through attitude [36]. As students gain more knowledge about CE, they will adopt more correct attitudes and behaviours, and vice versa [37]. Therefore, in order to change behaviors, we must begin by enhancing health education, addressing negative attitudes towards CE, and promoting positive behaviors.

The World Health Organization (WHO) has expanded health education from a focus on individual behaviour to a wide range of social and environmental interventions [38]. In the study, most of the students received health education through one of the means. Multimedia communication platforms were the most effective ways for students to acquire echinococcosis knowledge. Some students received health education through two or more than two means, indicating the importance of utilizing diverse educational channels for dissemination. Nowadays, the direction and degree of multimedia use in health education are gradually increasing, because it can cross the barriers of space and time, which makes most students find it convenient [39]. A study of oral health shows that the use of social media in oral health has become more popular than ever and will be used in more areas in the future [40]. Secondly, the way that students choose more is health education in schools. A study showed that when the school integrated health education into students' curriculum, it not only increased students' relevant knowledge, but also cultivated students' relevant skills [41]. However, Chinese undergraduate students have the least access to health knowledge from community hospitals, and community hospitals should strengthen the dissemination of health knowledge.

The students' KAP level of CE should be improved through school health education. First, it is necessary to introduce teaching materials on CE suitable for the acceptance level of college students of western areas in China. Second, school health educators should receive more specialized and comprehensive training to gain a deeper understanding of CE, which could bring CE knowledge into the classroom and focus on students with insufficient knowledge. Third, multimedia communication platforms will be used to regularly launch online health education promotion, so that college students can easily obtain and understand cystic echinococcosis-related knowledge. Fourthly, regular health education evaluation should be carried out to help policy makers or project planners to correct deficiencies in the current work. Beyond traditional school-based health education, community-based health education is also an important approach to improving students' KAP regarding echinococcosis. For example, tailored community education campaigns should be designed to address the specific needs and interests of university students, incorporating interactive sessions and relatable content that resonates with their daily lives. Partnerships with local community hospitals and community organisations can extend the reach and credibility of these initiatives and ensure access to expertise and resources. In addition, the use of community digital platforms, such as media platforms and mobile applications, can facilitate the dissemination of engaging and timely information.

However, there are some limitations to the study. Firstly, the data was obtained using a self-administered questionnaire. Secondly, the research was carried out in a higher education institution, and its findings may not be applicable to the broader population. Thirdly, assessing the honesty and seriousness of respondents in answering the questions is challenging. Fourthly, the majority of participants were Han Chinese students (93%), and nationality discrepancies may be a source of bias in the study. Therefore, follow-up research could increase the representativeness of the sample to reduce self-report bias and test the findings of this study. On the other hand, the next study could expand the sample to include not only more

than one higher education institution, but also people from different ethnic, cultural, and educational backgrounds.

## Conclusions

The Chinese students in western areas presented a low KAP level towards CE, but the KAP scores of female students and urban students were higher than those of male students and rural students, respectively. The passing rate of the college students of western areas in China for core knowledge of CE is 50%, but the overall excellence rate with a good level of knowledge is low, and most of the participants showed insufficient or poor knowledge of CE. It is necessary to strengthen the health education on CE for undergraduate students in western areas of China. Multimedia communication platforms and health education in schools were the most effective ways for students to acquire echinococcosis knowledge. Nonetheless, school health education remains a lengthy, gradual endeavor that demands increased focus and dedication.

## Supporting information

**S1 File.** **S1 Appendix**. STROBE Statement. **S2 Appendix**. Questionnaire of knowledge, attitudes, and practice of echinococcosis. **S3 Appendix**. The anonymized dataset of this study. (ZIP)

## Acknowledgments

The authors thank the students who participated in this research, and are deeply grateful to the anonymous reviewers and editors for their insightful questions, which improved the manuscript.

## Author contributions

**Conceptualization:** Xingming Ma.

**Data curation:** Yijie Xu, Congwei Shen.

**Formal analysis:** Chengkai Luo, Jiacheng Liu.

**Methodology:** Yijie Xu, Chengkai Luo, Jiacheng Liu, Congwei Shen.

**Software:** Chengkai Luo, Jiacheng Liu.

**Supervision:** Congwei Shen, Xingming Ma.

**Writing – original draft:** Yijie Xu.

**Writing – review & editing:** Xingming Ma.

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
