## [Decision Letter · Decision Letter 0]

20 Nov 2024

PONE-D-24-34434Evaluation of knowledge, attitude and practice towards echinococcosis among undergraduate students in ChinaPLOS ONE

Dear Dr. Ma,

Thank you for submitting your manuscript to PLOS ONE. After careful consideration, we feel that it has merit but does not fully meet PLOS ONE’s publication criteria as it currently stands. Therefore, we invite you to submit a revised version of the manuscript that addresses the points raised during the review process.

We look forward to receiving your revised manuscript.

Kind regards,

Mohamed Lounis

Academic Editor

PLOS ONE

**Journal Requirements:**

the Project of Sichuan University Student Ideological and Political Education Research Center (CSZ24049) and the Key Project for Education and Teaching Reform in Xihua University (XJJG2021040)

3. In the online submission form, you indicated that The analysed data is all in the manuscript, and the anonymized dataset of this study is available upon request from the corresponding author.

**Additional Editor Comments:**

The authors should be clear regarding the subject (echinococcis ganulosus or multilocularis). Tow parasites that do not share the same features. This point should be highlighted in all parts of the manuscript.

The discussion and the conclusion should be summarised.

Avoid the repretition of introductive setence (describing the objectives) in the results (lines 203-204, 09-210, 228-29, 243-44, 254-56, 271-75...).

Line 188-191: this point should be provided in the methods.

In the abstrat povide the total score of KAP.

Reviewers' comments:

Reviewer's Responses to Questions

**Comments to the Author**

1. Is the manuscript technically sound, and do the data support the conclusions?

Reviewer #1: Partly

Reviewer #2: Yes

2. Has the statistical analysis been performed appropriately and rigorously? 

Reviewer #1: Yes

Reviewer #2: Yes

3. Have the authors made all data underlying the findings in their manuscript fully available?

Reviewer #1: Yes

Reviewer #2: Yes

4. Is the manuscript presented in an intelligible fashion and written in standard English?

Reviewer #1: Yes

Reviewer #2: Yes

5. Review Comments to the Author

**Reviewer #1: ** The currents study aims to evaluate the status of Knowledge, Attitude, and Practices (KAP) regarding echinococcosis among undergraduate students from western areas of China. It also seeks to provide a basis for developing health education strategies for college students on echinococcosis prevention. There are several concerns with the methodology and findings of the study.

Major Concerns

1.The sample includes students with diverse backgrounds and varying levels of general information, introducing many confounding factors that could affect the reliability of the findings.

2.The study's findings cannot be generalized to the broader population of the region or even to other university students due to the specific sample.

3.The study confirms already expected findings, such as the relationship between education level and KAP, which does not add new information to the existing literature.

While the study provides some insights into the KAP of undergraduate students regarding echinococcosis, the heterogeneity of the sample and the lack of generalizability limit its impact. The findings largely confirm what is already known, offering little new data to the field.

**Reviewer #2: ** This is a very interesting, high-quality and well-constructed piece of research.

My only comment is that right from the introduction there is no differentiation between alveolar and cystic echinococcosis. Although these two diseases are endemic in China, their clinical picture and, above all, their parasite cycle are quite different: cystic echinococcosis (CE) has a domestic cycle and alveolar echinococcosis (AE) a wild cycle, which is why AE is a disease that is much more likely to be transmitted through food than through contact with a domestic dog.

Most of the studies already published on the subject focus on EC, which is really a disease that can be prevented in a number of ways, but mainly by controlling it in abattoirs so as not to perpetuate the cycle between intermediate and final hosts. As EA has a wild cycle, controlling the livestock chain is of no use and only deworming dogs and modifying behaviour are ways of preventing it.

Throughout your article, I suggest that you take this difference into account for greater clarity.

6. PLOS authors have the option to publish the peer review history of their article (what does this mean? ). If published, this will include your full peer review and any attached files.

**Do you want your identity to be public for this peer review?** For information about this choice, including consent withdrawal, please see our Privacy Policy .

Reviewer #1: **Yes: ** Bahador Sarkari

Reviewer #2: No

---

## [Author Response · Author response to Decision Letter 0]

26 Dec 2024

Dear Dr. reviewers1, reviewers2 and editors:

Thank you very much for your careful review and suggestions with regard to this manuscript. We considered carefully all the reviewers' comments and revised the manuscript meticulously, and made some changes in the manuscript according to the referees’ comments with tracked changes to highlight the revisions (red color). The main corrections in the paper (the revised manuscript) and the responds to the reviewer’ comments are uploaded the system.

We would like to thank you again for taking the time to review our manuscript.

The responses were attachment as following.

Dear Dr. reviewer 1:

Thank you very much for your careful review and suggestions with regard to this manuscript. We carefully revised the manuscript and made some changes in the manuscript with tracked changes to highlight the revisions (red color).

The currents study aims to evaluate the status of Knowledge, Attitude, and Practices (KAP) regarding echinococcosis among undergraduate students from western areas of China. It also seeks to provide a basis for developing health education strategies for college students on echinococcosis prevention. There are several concerns with the methodology and findings of the study.

Major Concerns

1.The sample includes students with diverse backgrounds and varying levels of general information, introducing many confounding factors that could affect the reliability of the findings.

[Answer] We greatly appreciate your insightful comment on the manuscript. Indeed, the sample in the study included students with diverse backgrounds and varying levels of general information. As much as possible, we try to control for confounding factors arising from the diverse backgrounds of the participants. Firstly, in the study, the students from western region were all those who took the China Unified College Entrance Examination and met the admission criteria for admission. Secondly, a cluster random sampling was used to recruit participants at the university. Thirdly, we have conducted an analysis on the sociodemographic variables, including gender, age, nationality, educational level, monthly living expenses, and residence. Fourth, we refered to similar literature studies to select samples, for example, Knowledge, attitudes, and practice with respect to antibiotic use among pharmacy students: a cross-sectional study (Eur Rev Med Pharmacol Sci, 2022;26(10):3408-3418); Knowledge, attitudes, and self-reported practices (KAP) towards hand hygiene in medical students versus the public. (Mwesigye P, Sekhon B, Punni A, et al. Ir J Med Sci. 2022;191(6):2797-2802. doi:10.1007/s11845-022-02918-x) etc.

2.The study's findings cannot be generalized to the broader population of the region or even to other university students due to the specific sample.

[Answer] We greatly appreciate your insightful comment on the manuscript. this is one of limitations to the study. We have added the limitations to the study and as following. In the future, we will further research with broader population of the region to enhance the generalizability of the findings.

…… However, there are some limitations to the study. Firstly, the data was obtained using a self-administered questionnaire. Secondly, the research was carried out in a higher education institution, its findings may not be applicable to the broader population. ……

3.The study confirms already expected findings, such as the relationship between education level and KAP, which does not add new information to the existing literature.

While the study provides some insights into the KAP of undergraduate students regarding echinococcosis, the heterogeneity of the sample and the lack of generalizability limit its impact. The findings largely confirm what is already known, offering little new data to the field.

[Answer] We would like to express our gratitude for your insightful comments on the manuscript. While our study does confirm the relationship between education level and KAP, which is consistent with existing literature, we believe that our research provides some insights into the echinococcosis awareness and prevention practices among a particular group of undergraduate students. Firstly, the subjects of this study are undergraduate students from western China in our university, which is also a region with a high incidence of echinococcosis. Therefore, the survey data here could reflect to some extent the KAP level of college students from western China about this disease. In addition, we also analyzed the methods and/or channels through which students in university received health education. This not only provides some insights of health education for the prevention and practice of this disease, but also it could offer some reference for the prevention and control of other epidemic diseases.

We sincerely thank you for your valuable suggestions. We would like to thank you again for taking the time to review our manuscript.

Many thanks again,

Best wishes.

Dear Dr. reviewer 2:

Thank you very much for your careful review and suggestions with regard to this manuscript. We carefully revised the manuscript and made some changes in the manuscript with tracked changes to highlight the revisions (red color).

This is a very interesting, high-quality and well-constructed piece of research. My only comment is that right from the introduction there is no differentiation between alveolar and cystic echinococcosis. Although these two diseases are endemic in China, their clinical picture and, above all, their parasite cycle are quite different: cystic echinococcosis (CE) has a domestic cycle and alveolar echinococcosis (AE) a wild cycle, which is why AE is a disease that is much more likely to be transmitted through food than through contact with a domestic dog.

Most of the studies already published on the subject focus on EC, which is really a disease that can be prevented in a number of ways, but mainly by controlling it in abattoirs so as not to perpetuate the cycle between intermediate and final hosts. As EA has a wild cycle, controlling the livestock chain is of no use and only deworming dogs and modifying behaviour are ways of preventing it.

Throughout your article, I suggest that you take this difference into account for greater clarity.

[Answer] We greatly appreciate your insightful comment on the manuscript. Indeed, the tow parasites (Echinococcis ganulosus or multilocularis) do not share the same characteristics, particularly in their life cycles and transmission cycle. This manuscript focuses on the evaluation of knowledge, attitude and practice towards cystic echinococcosis. We have revised the entire manuscript to focus on cystic echinococcosis which caused by echinococcus granulosus.

We sincerely thank you for your valuable suggestions. We would like to thank you again for taking the time to review our manuscript.

Many thanks again,

Best wishes.

Journal Requirements:

[Answer] Thank you for your guidance. We have carefully reviewed the PLOS ONE style requirements and have ensured that our manuscript, including file naming, adheres to these standards.

the Project of Sichuan University Student Ideological and Political Education Research Center (CSZ24049) and the Key Project for Education and Teaching Reform in Xihua University (XJJG2021040)

[Answer] We appreciate your attention to the financial disclosure. We confirm that the funders had no role in the study design, data collection and analysis, decision to publish, or preparation of the manuscript. We have included this statement in the manuscript and as following.

The authors declare that the study was funded by the Project of Sichuan University Student Ideological and Political Education Research Center (CSZ24049). The funders had no role in study design, data collection and analysis, decision to publish, or preparation of the manuscript.

3. In the online submission form, you indicated that The analysed data is all in the manuscript, and the anonymized dataset of this study is available upon request from the corresponding author.

[Answer] Thank you for your suggestion. We have uploaded all data as supplementary information, and all data underlying the findings described in the manuscript to be freely available to other researchers, and as following.

S1 Appendix. STROBE Statement.

S2 Appendix. Questionnaire of knowledge, attitudes, and practice of echinococcosis.

S3 Appendix. The anonymized dataset of this study.

[Answer] We acknowledge the requirement for an ORCID iD for the corresponding author. We have ensured that the corresponding author has an ORCID iD (0000-0002-1860-4764) and that it is validated in Editorial Manager.

[Answer] We have followed the Supporting Information guidelines and included captions for our Supporting Information files at the end of our manuscript. We have also updated all in-text citations to ensure they match the captions.

Additional Editor Comments:

The authors should be clear regarding the subject (echinococcis ganulosus or multilocularis). Tow parasites that do not share the same features. This point should be highlighted in all parts of the manuscript. The discussion and the conclusion should be summarised. Avoid the repretition of introductive setence (describing the objectives) in the results (lines 203-204, 09-210, 228-29, 243-44, 254-56, 271-75...). Line 188-191: this point should be provided in the methods. In the abstrat povide the total score of KAP.

[Answer] Thank you for your insightful comments. We have made the following revisions in the manuscript with tracked changes to highlight the revisions (red color).

The tow parasites (Echinococcis ganulosus or multilocularis) do not share the same characteristics. This manuscript focuses on cystic echinococcosis. We have revised the entire manuscript to emphasise cystic echinococcosis caused by Echinococcus granulosus. The discussion and conclusion sections have been summarized to provide a more concise overview of our findings. We have eliminated the repetition of introductive sentences in the results section, ensuring that each point is only mentioned once. The information mentioned in lines 188-191 has been incorporated into the methods section to maintain consistency and clarity. The abstract has been updated to include the total score of KAP.

We sincerely thank you for your valuable suggestions. We would like to thank you again for taking the time to review our manuscript.

Many thanks again,

Best wishes.

---

## [Decision Letter · Decision Letter 1]

31 Jan 2025

PONE-D-24-34434R1Evaluation of knowledge, attitude and practice towards cystic echinococcosis among undergraduate students in ChinaPLOS ONE

Dear Dr. Ma,

Thank you for submitting your manuscript to PLOS ONE. After careful consideration, we feel that it has merit but does not fully meet PLOS ONE’s publication criteria as it currently stands. Therefore, we invite you to submit a revised version of the manuscript that addresses the points raised during the review process.

**ACADEMIC EDITOR:**The authors made considerable efforts to improve the quality of the manuscript. Some remarks raised by reviewer 3 should be taken into consideration.

We look forward to receiving your revised manuscript.

Kind regards,

Mohamed Lounis

Academic Editor

PLOS ONE

Journal Requirements:

Reviewers' comments:

Reviewer's Responses to Questions

**Comments to the Author**

1. If the authors have adequately addressed your comments raised in a previous round of review and you feel that this manuscript is now acceptable for publication, you may indicate that here to bypass the “Comments to the Author” section, enter your conflict of interest statement in the “Confidential to Editor” section, and submit your "Accept" recommendation.

Reviewer #1: (No Response)

Reviewer #2: All comments have been addressed

Reviewer #3: All comments have been addressed

2. Is the manuscript technically sound, and do the data support the conclusions?

Reviewer #1: No

Reviewer #2: (No Response)

Reviewer #3: Yes

3. Has the statistical analysis been performed appropriately and rigorously? 

Reviewer #1: I Don't Know

Reviewer #2: (No Response)

Reviewer #3: Yes

4. Have the authors made all data underlying the findings in their manuscript fully available?

Reviewer #1: Yes

Reviewer #2: (No Response)

Reviewer #3: Yes

5. Is the manuscript presented in an intelligible fashion and written in standard English?

Reviewer #1: No

Reviewer #2: (No Response)

Reviewer #3: Yes

6. Review Comments to the Author

Reviewer #1: (No Response)

Reviewer #2: (No Response)

Reviewer #3: Evaluation of knowledge, attitude and practice towards cystic echinococcosis among undergraduate students in China

Some comments for improvement:

1. Details information about the questionnaire's development and validation process to establish its reliability and validity would be better.

2. Sampling procedure is somewhat obscure, and it should be clear and address potential selection bias.

3. Discuss ethical considerations, such as how informed consent was obtained and whether anonymity was ensured during data collection.

4. In the result and discussion section, The study provides detailed insights into KAP levels, but the discussion could be streamlined by summarizing repetitive details and focusing on key findings

5. While gender and urban-rural differences are highlighted, the reasons behind these disparities should be explored further to provide more depth

6. Multimedia and school-based education are identified as effective, but practical suggestions for improving community-based education would strengthen the recommendations.

7. The limitations are well-stated, but strategies for addressing these in future research (e.g., expanding the sample or reducing self-reporting biases) should be discussed.

7. PLOS authors have the option to publish the peer review history of their article (what does this mean? ). If published, this will include your full peer review and any attached files.

**Do you want your identity to be public for this peer review?** For information about this choice, including consent withdrawal, please see our Privacy Policy .

Reviewer #1: No

Reviewer #2: No

Reviewer #3: No

---

## [Author Response · Author response to Decision Letter 1]

28 Feb 2025

Dear Dr. Reviewer #3

Thank you very much for your careful review #3 and suggestions with regard to this manuscript. We considered carefully all the reviewers' comments and revised the manuscript meticulously, and made some changes in the manuscript according to the referees’ comments with tracked changes to highlight the revisions (red color).

Reviewer #3

1. Details information about the questionnaire's development and validation process to establish its reliability and validity would be better.

[Answer] Thank you for your suggestion. Before the formal questionnaire, we firstly discussed with a Chinese expert together to confirm that the version of questionnaire was culturally appropriate for Chinese students. Then, a pre-survey of questionnaire was conducted and 31 sample were collected, and then the Cronbach’s α coefficients of questionnaire were analyzed with 0.72 (knowledge, attitude, and practice scales were 0.84, 0.50, and 0.67, respectively). which indicated good internal consistency and could be used for formal trials.

These can be found on page 7, lines 149 to 156 of the Revised version and as following.

…… A questionnaire used to gather data on KAP was based on a literature review of previous studies [25,28,29]. The questionnaire included all 37 items, which were divided into five domains (S2 Appendix). Meanwhile, we discussed with a Chinese expert together to confirm that the version of questionnaire was culturally appropriate for Chinese students. Next, a pre-survey of questionnaire was conducted and 31 sample were collected, and then the Cronbach’s α coefficients were analyzed. The Cronbach's alpha coefficients of questionnaire were 0.72, which indicated a good internal consistency and could be used for formal trials.

2. Sampling procedure is somewhat obscure, and it should be clear and address potential selection bias.

[Answer]Thank you for your guidance. We have revised the content of the sampling procedure section of the manuscript.

Excel random function sampling process as following:

Column A: Enter the class numbers of the school, sorted in ascending order.

Column B: Put to the new class number from 1 to 300.

Column C: Input the RAND () function to generate random numbers in all new class number. Then 300 random numbers were showed in the column C. Next, ascending order, and then the sorting of the first fourteen classes.

There are 870 students in the fourteen classes. Among those students, 769 students from the provinces of Sichuan, Shaanxi, Gansu, Qinghai, Yunnan, Xizang, Ningxia, Xinjiang, Chongqing, Guizhou, and Guangxi. A total of 769 students were selected as the object sampling of study.

These can be found on page 6, lines 127 to 132 of the Revised version and as following.

…… A cluster random sampling approach was employed to recruit participants at Xihua University. Students were selected from three hundred eligible classes, and then fourteen classes were randomly selected using the excel random function sampling method. Next, 769 students from western China voluntarily participated in this cross-sectional study, regardless of grades and majors, including both medical and non-medical students.

3. Discuss ethical considerations, such as how informed consent was obtained and whether anonymity was ensured during data collection.

[Answer] Thank you for your suggestion. Before participating in the survey, all participants were required to complete an informed consent form. The form clearly explained the purpose of the study, the voluntary nature of participation, and the right to withdraw at any time without consequences. Only those who agree to participate the survey were to the questionnaire. This process ensured that all participants provided explicit consent prior to data collection. Both the informed consent form and the questionnaire were designed to be fully anonymous. No personally identifiable information (e.g., name, student ID, or contact details) was collected at any stage. Additionally, the data were aggregated and analyzed at the group level, further safeguarding participant privacy.

These can be found on page 7, lines 145 to 147 of the Revised version and as following.

…… Participants were guaranteed that their data would solely be utilized for research purposes. Both the informed consent form and the questionnaire were designed to be fully anonymous, and informed consent was obtained from all participants.

4. In the result and discussion section, The study provides detailed insights into KAP levels, but the discussion could be streamlined by summarizing repetitive details and focusing on key findings

[Answer] We greatly appreciate your insightful comment on the manuscript. We have revised parts of the discussion in the manuscript.

These can be found on page 12-13, lines 295 to 303 of the Revised version and as following.

…… Among the 724 participants, the findings revealed that less than one-fifth of undergraduate students had higher level knowledge, positive attitudes, and low-risk practices, indicating that the current overall CE health literacy needs to be improved. Urban students exhibited higher KAP scores compared to rural counterparts, and disparities in knowledge, attitude, and practices based on gender and residence. Multimedia communication platforms and school health education were the most effective ways for students to acquire echinococcosis knowledge, indicating the importance of utilizing diverse educational channels for dissemination.

5. While gender and urban-rural differences are highlighted, the reasons behind these disparities should be explored further to provide more depth

[Answer] We greatly appreciate your insightful comment on the manuscript. According to the data obtained by the experiment, we have revised parts of the discussion in the manuscript.

These can be found on page 14, lines 337 to 357 of the Revised version and as following.

…… Female and urban students displayed better practices than males and rural students, underlining gender and residence disparities in CE management behaviors. Females were at higher risk of echinococcosis infection than males, mainly because they were more involved in most family chores and activities, such as feeding dogs and preparing food for their families [5]. Women are often seen as family health managers, and society expects them to actively seek disease prevention information, while men may ignore such knowledge through role labels [5]. To reduce the risk of CE, female students tended to acquire more health knowledge about the echinococcosis disease and adopt health behaviors. Similarly, urban students, benefiting from greater access to educational opportunities and resources, were more likely to gain comprehensive knowledge about CE and implement preventive health practices. Similarly, urban students, benefiting from greater access to educational opportunities and resources, were more likely to gain comprehensive knowledge about CE and implement preventive health practices. There are also differences in the health ecology of communities. Urban communities tend to have better food quarantine systems and medical advice channels to support preventive behaviour. Such resources are difficult to put into practice in rural areas, even where knowledge exists. In addition, rural families may put disease prevention on the backburner due to economic pressures, which indirectly affects students' value perception of health behaviours.

6. Multimedia and school-based education are identified as effective, but practical suggestions for improving community-based education would strengthen the recommendations.

[Answer] We greatly appreciate your insightful comment on the manuscript. We have revised parts of the discussion in the manuscript.

These can be found on page 16, lines 390 to 408 of the Revised version and as following.

…… The students’ KAP level of CE should be improved through school health education. First, it is necessary to introduce teaching materials on CE suitable for the acceptance level of college students of western areas in China. Second, school health educators should receive more specialized and comprehensive training to gain a deeper understanding of CE, which could bring CE knowledge into the classroom and focus on students with insufficient knowledge. Third, multimedia communication platforms will be used to regularly launch online health education promotion, so that college students can easily obtain and understand cystic echinococcosis-related knowledge. Fourthly, regular health education evaluation should be carried out to help policy makers or project planners to correct deficiencies in the current work. Beyond traditional school-based health education, community-based health education is also an important approach to improving students' KAP regarding echinococcosis. For example, tailored community education campaigns should be designed to address the specific needs and interests of university students, incorporating interactive sessions and relatable content that resonates with their daily lives. Partnerships with local community hospitals and community organisations can extend the reach and credibility of these initiatives and ensure access to expertise and resources. In addition, the use of community digital platforms, such as media platforms and mobile applications, can facilitate the dissemination of engaging and timely information.

7. The limitations are well-stated, but strategies for addressing these in future research (e.g., expanding the sample or reducing self-reporting biases) should be discussed.

[Answer] Thank you for your insightful comments. We have revised parts of the limitations in the manuscript.

These can be found on page 16-17, lines 409 to 418 of the Revised version and as following.

…… However, there are some limitations to the study. Firstly, the data was obtained using a self-administered questionnaire. Secondly, the research was carried out in a higher education institution, and its findings may not be applicable to the broader population. Thirdly, assessing the honesty and seriousness of respondents in answering the questions is challenging. Fourthly, the majority of participants were Han Chinese students (93%), and nationality discrepancies may be a source of bias in the study. Therefore, follow-up research could increase the representativeness of the sample to reduce self-report bias and test the findings of this study. On the other hand，the next study could expand the sample to include not only more than one higher education institution，but also people from different ethnic，cultural，and educational backgrounds.

We sincerely thank you for your valuable suggestions. We would like to thank you again for taking the time to review our manuscript.

Many thanks again,

Best wishes.

---

## [Editor Report · Decision Letter 2]

5 Mar 2025

Evaluation of knowledge, attitude and practice towards cystic echinococcosis among undergraduate students in China

PONE-D-24-34434R2

Dear Dr. Ma,

We’re pleased to inform you that your manuscript has been judged scientifically suitable for publication and will be formally accepted for publication once it meets all outstanding technical requirements.

Kind regards,

Mohamed Lounis

Academic Editor

PLOS ONE

Additional Editor Comments (optional):

The authors have addressed all comments provided by the reviewers.
---

## [Editor Report · Acceptance letter]

PONE-D-24-34434R2

PLOS ONE

Dear Dr. Ma,

I'm pleased to inform you that your manuscript has been deemed suitable for publication in PLOS ONE. Congratulations! Your manuscript is now being handed over to our production team.

Kind regards,

on behalf of

Dr. Mohamed Lounis

Academic Editor

PLOS ONE